# Influenza in Liver and Kidney Transplant Recipients: Incidence and Outcomes

Nicoline Stender Arentoft,[a] Dina Leth Møller,[a] Andreas Delhbæk Knudsen,[a] Ranya Abdulovski,[a] Nikolai Kirkby,[b] Søren Schwartz Sørensen,[c,d] Allan Rasmussen,[e] Susanne Dam Nielsen[a,d,e]

[a]Viro-immunology Research Unit, Department of Infectious Diseases 8632, Rigshospitalet, University of Copenhagen, Copenhagen, Denmark

[b]Department of Clinical Microbiology, Rigshospitalet, University of Copenhagen, Copenhagen, Denmark

[c]Department of Nephrology, Rigshospitalet, University of Copenhagen, Copenhagen, Denmark

[d]Department of Clinical Medicine, University of Copenhagen, Copenhagen, Denmark

[e]Department of Surgical Gastroenterology and Transplantation, Rigshospitalet, University of Copenhagen, Copenhagen, Denmark

**ABSTRACT** Influenza is a common respiratory tract infection in solid organ transplant (SOT) recipients. We aimed to investigate the incidence, risk factors, and complications of influenza in a large cohort of kidney and liver transplant recipients over 10 consecutive seasons. We conducted a retrospective study, including 378 liver and 683 kidney transplant recipients who were transplanted from January 1, 2010, to October 1, 2019. The data on influenza were retrieved from MiBa, which is a nationwide database that contains all of the microbiology results in Denmark. Clinical data were retrieved from patient records. Incidence rates and cumulative incidences were calculated, and risk factors were investigated using time-updated Cox proportional hazards models. The cumulative incidence of influenza in the first 5 years posttransplantation was 6.3% (95% CI: 4.7 to 7.9%). Of the 84 influenza positive recipients, 63.1% had influenza A, 65.5% were treated with oseltamivir, 65.5% were hospitalized, and 16.7% developed pneumonia. There were no significant differences in outcomes when comparing patients with influenza A and B. We found no significant effect of same-season influenza vaccination, sex, age, or comorbidities on the risk of acquiring influenza. The incidence of influenza in kidney and liver recipients is high, and 65.5% of infected transplant recipients required hospitalization. We were not able to confirm a reduction in influenza incidence or in the risk of complications associated with vaccination.

**IMPORTANCE** Influenza is a common respiratory virus in solid organ transplant recipients that may have severe complications, including pneumonia and hospitalization. This study investigates the incidence, risk factors, and complications of influenza in a Danish cohort of kidney and liver transplant recipients over 10 consecutive influenza seasons. The study shows a high incidence of influenza and a high frequency of both pneumonia and hospitalization. This emphasizes the importance of continuous focus on influenza in this vulnerable group. During the COVID-19 pandemic, the incidence of influenza has been low due to COVID-related restrictions, and immunity may have waned. However, as most countries have now opened up, the incidence of influenza is expected to be high this season.

**KEYWORDS** influenza, kidney transplantation, liver transplantation, medical outcomes, risk factors, transplantation, vaccines

Address correspondence to Susanne Dam Nielsen, sdn@dadlnet.dk.

The authors declare a conflict of interest. N.S.A., D.L.M., R.A., N.K., S.S.S., and A.R. did not report any conflict of interest. A.D.K. received a grant from The Danish Heart Foundation and a grant from the European Commission not related to this work. S.D.N. received a grant from the Novo Nordic Foundation.

Due to the immunosuppressive therapy, solid organ transplant (SOT) recipients are at increased risk of infections, compared to immunocompetent individuals (1). Influenza is a common respiratory infection in kidney and liver recipients, and kidney transplant recipients have a 5-fold higher risk of influenza, compared to the background population (2, 3).

A SOT recipient with influenza may have a clinical presentation that is similar to that of an immunocompetent individual (4). However, influenza often has a more

**TABLE 1** Characteristics

| Characteristics | All (n = 1,061) | Kidney transplanted recipients (n = 683) | Liver transplanted recipients (n = 378) |
|---|---|---|---|
| Age at transplantation, yr, median (range) | 50.8 (18.0 to 83.5) | 50.9 (19.0 to 83.5) | 50.2 (18.0 to 73.8) |
| Male sex, n (%) | 646 (61.0%) | 428 (62.7%) | 218 (57.7%) |
| Patients with ≥1 comorbidity at transplantation, n (%) | | | |
| Diabetes mellitus type I or II, n (%) | 180 (17.0%) | 121 (17.7%) | 59 (15.6%) |
| Cardiovascular disease, n (%) | 689 (64.9%) | 611 (89.5%) | 78 (20.6%) |
| Chronic lung disease, n (%) | 102 (9.6%) | 66 (9.7%) | 36 (9.5%) |
| Dead, n (%) | 212 (20.0%) | 134 (19.6%) | 78 (20.6%) |
| Rejection, n (%) | 284 (26.8%) | 206 (30.2%) | 78 (20.6%) |
| Influenza vaccinated in any season, n (%) | 713 (67.2%) | 475 (69.5%) | 238 (63.0%) |
| Influenza vaccinated in 2010, n (%) | 141 (13.3%) | 97 (14.2%) | 44 (11.6%) |
| Influenza vaccinated in 2011, n (%) | 161 (15.2%) | 112 (16.4%) | 49 (13.0%) |
| Influenza vaccinated in 2012, n (%) | 175 (16.5%) | 121 (17.7%) | 54 (14.3%) |
| Influenza vaccinated in 2013, n (%) | 212 (20.0%) | 139 (20.4%) | 73 (19.3%) |
| Influenza vaccinated in 2014, n (%) | 192 (18.1%) | 128 (18.7%) | 64 (16.9%) |
| Influenza vaccinated in 2015, n (%) | 235 (22.1%) | 155 (22.7%) | 80 (21.2%) |
| Influenza vaccinated in 2016, n (%) | 306 (28.8%) | 196 (28.7%) | 110 (29.1%) |
| Influenza vaccinated in 2017, n (%) | 331 (31.2%) | 213 (31.2%) | 118 (31.2%) |
| Influenza vaccinated in 2018, n (%) | 370 (34.9%) | 237 (34.7%) | 133 (35.2%) |
| Influenza vaccinated in 2019, n (%) | 370 (34.9%) | 234 (34.3%) | 136 (36.0%) |

aggressive course in SOT recipients, and complications, including pneumonia and bacterial or fungal coinfections, are common (5, 6). Hospitalization due to influenza in kidney transplant recipients is fourfold higher than is observed in the general population, and a mortality reaching 5% has been reported (3).

In SOT recipients, seasonal influenza vaccination entails a lower risk of complications, and annual influenza vaccination is recommended (7, 8). However, adherence to the vaccination programs in SOT recipients is often low, with a coverage of 38% to 51% (7, 9, 10). Importantly, the early initiation of antiviral therapy is associated with a lower risk of complications (4). In contrast, older age, the recent use of a high-dose corticosteroid, and recent episodes of rejection have been suggested to be risk factors that are associated with complications to influenza in SOT recipients (3, 5).

In this study of a large cohort of kidney and liver transplant recipients in Denmark, we investigated the incidence of influenza in 10 consecutive seasons, from the 2010/2011 season to the 2019/2020 season. Furthermore, we describe the risk factors that are associated with the acquisition of influenza as well as complications related to influenza.

## RESULTS

Between January 1, 2010, and October 1, 2019, 683 kidney and 378 liver transplant recipients were included in the study (Table 1). The median age at the time of transplantation was 51 years (range 18 to 84), and 61% of the cohort were men. The distribution of comorbidities is shown in Table 1. A total of 386 episodes of acute rejection were observed in 294 recipients (27.7%).

**Influenza vaccination.** A total of 713 recipients (67%) were influenza vaccinated in at least one of the 10 seasons, and 124 (11.7%), 100 (9.4%), and 34 (3.2%) recipients were vaccinated in two, three, and all seasons, respectively. Only 19 (22.6%) received an influenza vaccination in the same season as their influenza infection. The distribution of vaccinated recipients was comparable in the two organ groups (69.5% versus 63.0%).

**Influenza.** The median follow-up was 5.6 years (interquartile range [IQR] of 3.4 to 8.3), with a total follow-up of 6,152 person-years in the cohort. 81 recipients (7.6%) were diagnosed with influenza during the follow-up. Of these, 3 recipients had influenza twice in different seasons, resulting in a total of 84 influenza infections in the cohort. The median time to infection was 824 days posttransplantation (IQR of 251 to 1,930 days). There were significantly more influenza infections in the kidney recipients than in the liver recipients

**TABLE 2** Influenza positive recipients versus influenza negative recipients

| Characteristics | Influenza positive recipients[a] | Influenza negative recipients | P values |
|---|---|---|---|
| No. of patients, n | 81 | 980 | |
| Age at transplantation, yr, median (range) | 49.7 (19.4 to 71.2) | 50.9 (18.0 to 83.5) | P = 0.6 |
| Male sex, n (%) | 46 (56.8%) | 600 (61.3%) | P = 0.6 |
| Transplanted organ | | | |
| Kidney | 64 (79.0%) | 617 (63.1%) | P = 0.008 |
| Liver | 17 (21.0%) | 361 (36.9%) | |
| Influenza vaccination in any season | 56 (69.1%) | 655 (67.0%) | P = 0.9 |
| Patients with ≥1 comorbidity, n (%) | | | |
| Diabetes mellitus type I and II, n (%) | 14 (17.3%) | 166 (17.0%) | P = 0.9 |
| Chronic heart disease, n (%) | 61 (75.3%) | 626 (64.0%) | P = 0.1 |
| Chronic lung disease, n (%) | 9 (11.1%) | 93 (9.5%) | P = 0.9 |
| Rejection, n (%) | 27 (33.3%) | 255 (26.1%) | P = 0.2 |
| Dead, n (%) | 17 (21.0%) | 194 (19.8%) | P = 0.9 |

[a]Three recipients had influenza in two different seasons.

(79.0% versus 63.1%, P = 0.008). There were no other significant differences in clinical characteristics or in the number of influenza vaccinations in any season between the recipients with influenza, compared to the recipients without influenza (Table 2).

**Influenza incidence rate and cumulative incidence.** The incidence rate of influenza in each season from the 2010/2011 season to the 2019/2020 season ranged between 0 (0.0 to 1.6) and 5.4 (95% confidence interval [CI] of 1.8 to 12.9) per 1,000 person-months at risk (Fig. S1; Table S2). There were no significant differences in the incidence rates between the recipients who had received same-season influenza vaccination and the recipients who had not (Table S2).

The cumulative incidences of the first influenza infection in the first year and the first 5 years posttransplantation were 2.5% (95% CI:1.6 to 3.5%) and 6.3% (95% CI: 4.7 to 7.9%), respectively (Fig. 1). The kidney transplant recipients had a higher cumulative

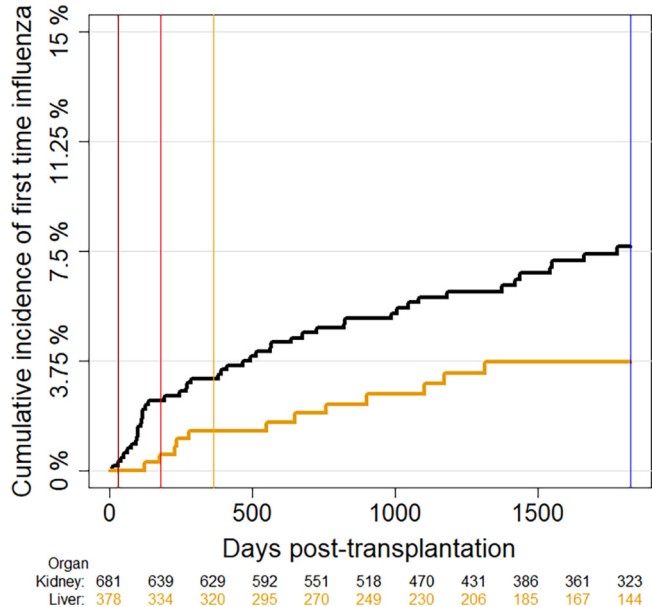

**FIG 1** Vertical lines mark 1 month (maroon), 6 months (red), 1 year (yellow), and 5 years (blue) posttransplantation. The cumulative incidences are shown for kidney transplant recipients (black line) and liver transplant recipients (yellow line).

**TABLE 3** Risk factors

| Risk factor | Unadjusted (95% CI) | Adjusted for age and sex (95% CI) | Adjusted for age, sex and having ≥1 comorbidity (95% CI) |
|---|---|---|---|
| Age | 1.0 (0.98 to 1.0), P = 0.8 | 1.0 (0.98 to 1.0), P = 0.8 | 1.0 (0.98 to 1.0), P = 0.8 |
| Sex | 0.9 (0.6 to 1.3), P = 0.5 | 0.9 (0.6 to 1.3), P = 0.5 | 0.9 (0.6 to 1.3), P = 0.5 |
| Comorbidity | 1.4 (0.8 to 2.6) P = 0.3 | 1.5 (0.8 to 2.7) P = 0.2 | 1.5 (0.8 to 2.7) P = 0.2 |
| Same-season influenza vaccine | 1.3 (0.8 to 2.2), P = 0.3 | 1.4 (0.8 to 2.2), P = 0.2 | 1.4 (0.8 to 2.3) P = 0.2 |

incidence of influenza than did the liver recipients (7.7% [95% CI 5.5 to 9.8%] versus 3.7% [95% CI 1.6 to 5.8%], $P = 0.02$).

**Risk factors for influenza.** Risk factors associated with being diagnosed with influenza were tested in a time-updated Cox proportional hazards model. Age, sex, having at least one comorbid disease, and same-season influenza vaccination were not associated with the risk of influenza (Table 3). In unadjusted and adjusted analyses, the risk of acquiring influenza had hazard ratios for same-season vaccination of 1.3 (95% CI 0.8 to 2.2, $P = 0.3$) and 1.4 (95% CI 0.8 to 2.3, $P = 0.2$), respectively.

**Outcome after influenza infection.** Of the 84 influenza infections, 66 (78.6%) and 18 (21.4%) were found in kidney and liver recipients, respectively (Table 4). Of the influenza positive recipients, 55 of the 84 (65.5%) were treated with oseltamivir, 55 (65.5%) were hospitalized, and 14 recipients (16.7%) developed pneumonia. The clinical symptoms of the remaining 70 recipients (83.3%) in relation to influenza can be seen in Table S3. 6 (7.1%) recipients were admitted to an intensive care unit (ICU) and had supportive mechanical ventilation within 30 days of their influenza infections. Only 1 (1.2%) recipient died within 30 days of a positive influenza polymerase chain reaction (PCR). There were no significant differences in outcomes when comparing kidney and liver recipients (Table 4), nor did we find any differences in outcomes when comparing influenza positive recipients who had received same-season vaccination with those who had not (Table S4). Pneumonia was numerically more common in the recipients who had not been vaccinated in the same season, but the finding was not statistically significant (10.5% versus 18.5%, $P = 0.7$). When comparing outcomes for patients with influenza A and influenza B, there were no significant differences, although hospitalization tended to be more common in patients with influenza A than in patients with influenza B (73.6% versus 51.6%, $P = 0.07$) (Table S5).

## DISCUSSION

This retrospective cohort study of kidney and liver transplant recipients investigated the cumulative incidence of influenza in the first 5 years posttransplantation, showing a cumulative incidence that continues to rise over time and a higher incidence in kidney transplant recipients, compared to liver transplant recipients. Complications and risk factors

**TABLE 4** Outcomes after influenza infection in kidney and liver transplanted recipients

| Outcome | All influenza infections (n =84) | Influenza infections in kidney transplanted recipients (n = 66 [79%]) | Influenza infections in liver transplanted recipients(n = 18 [21%]) | P values |
|---|---|---|---|---|
| Time from tx to influenza, days, median (IQ range) | 824 (251.3 to 1929.8) | 792 (209.5 to 2059) | 1002.5 (345.3 to 1769.3) | P = 1 |
| Same-season influenza vaccination | 19 (22.6%) | 17 (25.8%) | 2 (11.1%) | P = 0.4 |
| Treated with oseltamivir (%) | 55 (65.5%) | 45 (68.2%) | 10 (55.6%) | P = 0.7 |
| Pneumonia (%) | 14 (16.7%) | 13 (19.7%) | 1 (5.6%) | P = 0.3 |
| Hospital admission (%) | 55 (65.5%) | 44 (66.7%) | 11 (61.1%) | P = 1 |
| ICU admission (%) | 6 (7.1%) | 5 (7.6%) | 1 (5.6%) | P = 1 |
| Mechanical ventilation (%) | 5 (6.0%) | 4 (6.1%) | 1 (5.6%) | P = 1 |
| Death, 30 days all-cause mortality (%) | 1 (1.2%) | 1 (1.5%) | 0 (0%) | P = 1 |
| | | | | |
| Influenza type (%) | | | | |
| A | 53 (63.1%) | 44 (66.7%) | 9 (50%) | P = 0.7 |
| B | 31 (36.9%) | 22 (33.3%) | 9 (50%) | P = 0.5 |

associated with influenza were studied with no significant effect of vaccination in the same season as influenza and no association with any other potential risk factors. Hospitalization and pneumonia were common complications to influenza in both kidney and liver recipients, and the frequency was not lower in recipients who had received same-season vaccination.

We found cumulative incidences of influenza of 2.5% and 6.3% in the first year and the first 5 years posttransplantation, respectively. A recent Swiss study, which included 3,294 SOT recipients, found a similar cumulative incidence of influenza in nonlung transplant recipients of 7.5% in the first 7.5 years posttransplantation (2). In general, SOT recipients are most susceptible to infections in the early posttransplantation phase due to the high immunosuppression (1). However, the cumulative incidences in our study showed influenza in both early and later posttransplantation periods, which may be explained by influenza being a community-acquired viral infection that poses a persistent risk of infection every year in the influenza season, which is unrelated to immunosuppression. These results highlight the importance of continuous awareness of influenza, posttransplantation.

Our study showed an incidence rate between 0 and 5.4 per 1,000 person months in 10 consecutive influenza seasons. An American study that included 3,569 SOT recipients found a similar incidence rate of influenza that varied between 2.8 and 4.3 per 1,000 person-years in lung, liver, and kidney transplant recipients (11). This highlights the importance of including multiple influenza seasons when investigating influenza epidemiology. We found zero influenza events in the 2011/2012 season, in accordance with the general population in Denmark, in which the number of patients with influenza-like symptoms was at the lowest level since surveillance started in 1994 (12). Likewise, the high incidence rate of influenza in the 2017/2018 season reflected the unusually long influenza season with high activity in the general population in Denmark (13).

In contrast to our expectations, same-season influenza vaccination was not associated with a lower risk of acquiring influenza. Likewise, we did not find evidence of fewer complications to influenza in recipients who had received same-season vaccination. There could be several possible reasons for this observation. First, studies of standard-dose influenza vaccination have shown a lower overall antibody response in SOT, compared to healthy controls (7). Furthermore, the strains in the yearly vaccine may not protect against the circulating strains (14). Lastly, our cohort may have been underpowered due to the various protection of the yearly vaccination. However, annual vaccination is recommended, as influenza vaccination is considered to be safe in SOT recipients (4, 7), and vaccination has been associated with less severe disease in several studies, including a recent Danish study from our group which found reduced risks of hospitalization and mortality in vaccinated recipients, following an influenza infection (8). Our study did not show an association between age, sex, comorbidities, and the risk of influenza. Only a few other studies of risk factors of influenza in SOT have been done, including a study from Finland of kidney transplant recipients, which found the time period in which the transplantation had been performed to be the only significant risk factor (3).

Our study found that influenza infections were more common in kidney transplant recipients than in liver transplant recipients, but there was no difference in complications associated with influenza infections between the organ groups. Hospitalization was seen in 65.5% of the kidney and liver transplant recipients with influenza, and 7.1% were admitted to an ICU. In a recent study from Canada that included 443 SOT recipients with influenza, 69.3% of the cohort were hospitalized, and 7.8% needed mechanical ventilation (4). Furthermore, a Swiss study of 186 influenza infections in SOT recipients found that 42.9% of the patients needed hospital admission and that 4.3% required mechanical ventilation (2). In Finland, kidney recipients have a fourfold higher risk of hospitalization due to influenza, compared to the general population, thereby emphasizing the increased risk associated with influenza in SOT recipients (3).

The strengths of our study include its large, well-described cohort that has a long, complete follow-up across 10 consecutive influenza seasons, based on Scandinavian registers as well as complete coverage of influenza test results. However, our study also had possible limitations. Information on vaccination became mandatory on November 15, 2015, but

when comparing vaccination numbers across seasons in our cohort, it does not seem like fewer vaccines were registered before 2015. Our database does not include data on immunosuppression, apart from treatment for rejection. Despite our complete coverage of influenza PCR results in Denmark, there is no routine influenza testing of transplant recipients. Therefore, patients with mild symptoms, such as vaccinated patients, are unlikely to be tested, and this patient group may be underrepresented in this study. Last, the number of influenza infections in our cohort limits the power of our statistical analyses.

In summary, this study provides new knowledge on the epidemiology of influenza in kidney and liver transplant recipients. Incidence rates of influenza in kidney and liver recipients fluctuates, following the influenza activity in the general population. The cumulative incidence in our study demonstrates the importance of continuous awareness of influenza in kidney and liver recipients. Influenza often leads to pneumonia and hospitalization in kidney and liver transplant recipients, thereby highlighting the severe consequences of influenza in these groups. We were not able to confirm a reduction in the risk of influenza or in the risk of complications associated with same-season vaccination.

## MATERIALS AND METHODS

**Study design and participants.** We present data from The Knowledge Center for Transplantation database at Rigshospitalet, Copenhagen, Denmark. The database includes all first-time liver and kidney transplant recipients who were transplanted at Rigshospitalet from January 1, 2010, to October, 2019. All liver transplantations in Denmark are performed at Rigshospitalet, whereas Denmark has 3 centers for kidney transplantation. The end of the follow-up was October 1, 2020, allowing for at least 1 year of follow-up for all recipients.

Danish citizens are registered using civil registration (CPR) numbers. A CPR number is a unique, personal number. All data were collected retrospectively from patient records and national databases. Data on influenza results were collected from the Danish microbiology database (MiBa), which contains data on all of the microbiology from all of the Departments of Clinical Microbiology in Denmark, with complete coverage since 2010 (15). Information about influenza vaccination was retrieved from the national Danish Vaccination Registry (DDV), which contains all of the vaccination data from primary care, hospital, and vaccination clinic settings. On November 15, 2015, it became mandatory for health care professionals to register all vaccinations in DDV. Furthermore, the registry contains voluntary vaccination information from before 2015. We included information about influenza vaccinations that were administered both before and after November 15, 2015. The influenza vaccines that are used in Denmark are inactivated and contain 3 or 4 strains. The used strains were as recommended by the World Health Organization (WHO) (Table S1) (14). We collected pretransplantation characteristics, including the age at transplantation, sex, date of transplantation, and comorbidities at the time of transplantation (diabetes mellitus type I and II, hypertension, chronic heart failure, ischemic heart disease, chronic obstructive pulmonary disease, and asthma). The posttransplant characteristics that were collected were acute organ rejections, influenza vaccination status, rejection treatment, and influenza test results. We collected information about influenza-specific outcomes, including antiviral treatment, pneumonia, hospital admission, admission to an ICU, mechanical ventilation, and death.

The retrieval of data was approved by the Centre for Regional Development (R-20051155). The requirement for approval by an ethics committee was waived in accordance with Danish law, which stipulates that informed consent or approval by an ethics committee is not needed for studies based on retrospective, aanonymized data.

**Laboratory influenza analysis.** All 10 national Departments of Clinical Microbiology in Denmark contributed to the MiBa database during the study period. Generally, samples were collected as oropharyngeal swabs, nasopharyngeal swabs, or a combination of both. Other sample materials used for influenza testing are bronchoalveolar lavage fluid and nasopharyngeal/nasal aspirate. Samples were analyzed via a real-time reverse transcription polymerase chain reaction (RT-PCR) analysis, using in-house or commercial tests at a central lab or point-of-care tests in clinical wards. Multiplex tests, including at least influenza A, influenza B, and an internal control, were used at all sites.

**Variable definitions.** One influenza season was defined as the period from October 1 to April 30 in two consecutive calendar years (e.g., October 1, 2010 to April 30, 2011), resulting in a maximum of 211 days at risk per influenza season. Each recipient was at risk of influenza infection in each influenza season, posttransplantation. Having influenza was defined as a positive PCR test for influenza A or B.

In the liver transplant recipients, acute rejection was defined as acute rejection that was treated with 1 g of methylprednisolone for 3 to 5 days. In the kidney transplant recipients, acute rejection was defined as acute rejection that was treated with 250 mg or 500 mg of methylprednisolone for 3 days.

**Outcome definitions.** The investigated outcomes were antiviral treatment, pneumonia, hospitalization, ICU admission, mechanical ventilation, and death. Antiviral treatment was defined as treatment with the neuraminidase inhibitor oseltamivir that was initiated between 7 days before and 7 days after a confirmed influenza infection. Pneumonia was defined as infiltrates on a chest X-ray together with a radiology report that was consistent with pneumonia. Hospitalization, ICU admission, and mechanical ventilation 5 days prior to an influenza infection and up to 30 days after an infection were included as outcomes of influenza. These outcomes were assessed only for individuals with an influenza infection.

**Statistical analyses.** Comparisons were calculated using the Mann-Whitney U test for the continuous variables and the chi-square test or Fisher's exact test for the categorical variables. The incidence rates (IR) of the first influenza infection per season were calculated as the numbers of recipients with an influenza infection per person month at risk for each influenza season. We calculated 95% confidence intervals using Byar's approximation to the Poisson distribution. The 5-year cumulative incidence of influenza posttransplantation was calculated using the Aalen-Johansen estimator, with death and retransplantation as competing risks. We used Gray's test to compare the cumulative incidences between the kidney and liver transplanted recipients (16). The risk factors for influenza infection were investigated in a time-updated multivariable Cox proportional hazards model with acute rejection and vaccination as time-updated covariates. The model was adjusted for age, sex, and pretransplant comorbidities. All of the analyses were conducted using the R statistical software package, version 3.6.1, with the "survival", "ggplot2", and "cmprsk" packages (17).

## SUPPLEMENTAL MATERIAL

Supplemental material is available online only.

**SUPPLEMENTAL FILE 1**, PDF file, 0.1 MB.

## ACKNOWLEDGMENTS

N.S.A., D.L.M., S.S.S., A.D.K., A.R., and S.D.N. designed the study. All authors collected the data. N.S.A. and D.L.M. performed the statistical analyses. N.S.A., D.L.M., and S.D.N. wrote the manuscript. All authors revised and commented on the manuscript. All authors read and approved the final version of the manuscript.

This work received funding from the Novo Nordic Foundation.

N.S.A., D.L.M, R.A., N.K., S.S.S., and A.R. have no conflicts of interest to report. A.D.K. received a grant from The Danish Heart Foundation and a grant from the European Commission not related to this work. S.D.N. received a grant from the Novo Nordic Foundation.

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
