## [Reviewer comments · Microbiology Spectrum]

Microbiology Spectrum

Influenza in Liver and Kidney Transplant Recipients: Incidence and Outcomes

Nicoline Arentoft, Dina Møller, Andreas Knudsen, Ranya Abdulovski, Nikolai Kirkby, Søren Sørensen, Allan Rasmussen, and Susanne Nielsen

Corresponding Author(s): Susanne Nielsen, Copenhagen University Hospital, Rigshospitalet

Review Timeline:

Submission Date:	August 17, 2022
Editorial Decision:	December 23, 2022
Revision Received:	February 1, 2023
Accepted:	March 9, 2023

Editor: Daniel Perez

Reviewer(s): Disclosure of reviewer identity is with reference to reviewer comments included in decision letter(s). The following individuals involved in review of your submission have agreed to reveal their identity: Michael Owusu (Reviewer #1)

Transaction Report:

DOI: <https://doi.org/10.1128/spectrum.03226-22>

December 23, 2022

Prof. Susanne Dam Nielsen
Copenhagen University Hospital, Rigshospitalet
Viro-immunology Research Unit - Department of Infectious Disease
Blegdamsvej 9
Copenhagen
Denmark

Re: Spectrum03226-22 (Influenza in Liver and Kidney Transplant Recipients: Incidence and Outcomes)

Dear Prof. Susanne Dam Nielsen:

Thank you for submitting your manuscript to Microbiology Spectrum. Please note that we had a very hard time identifying reviewers to agree to review your manuscript. In order to not delay any further the review process you will find below the comments from a single reviewer. When submitting the revised version of your paper, please provide (1) point-by-point responses to the issues raised by the reviewers as file type "Response to Reviewers," not in your cover letter, and (2) a PDF file that indicates the changes from the original submission (by highlighting or underlining the changes) as file type "Marked Up Manuscript - For Review Only". Please use this link to submit your revised manuscript - we strongly recommend that you submit your paper within the next 60 days or reach out to me. Detailed instructions on submitting your revised paper are below.

Link Not Available

Sincerely,

Daniel Perez

Journals Department
Reviewer comments:

Reviewer #1 (Comments for the Author):

Influenza is a respiratory virus that is associated with morbidity and mortality across the world. The incidence, risk factors and complications among cohorts of kidney and liver transplant recipients has not been extensively studied.

This study is therefore useful and likely to inform policy and stakeholders in making key decisions about these infections.

Authors should respond to the comments below:

line 58: Authors should state the number of positives relating to the second sentence. Eg: "Of the xxx influenza positive

recipients, 65.5% were

Line 108: "Citizens" should be "citizens"

Line 225-227: Authors should recheck the statement again. It appears positive cases were 84. Fourteen (14) recipients developed pneumonia so were the rest asymptomatic?

Line 296-298: Authors mentioned that some vaccinations administered could not be included. Authors should indicate the number of number of vaccinations that were not included.

General comments

Table 4: Although authors included flu A and B results, this was not described in the text. Flu A is much more severe than Flu B so it will be interesting to discuss this. The abstract should as well capture Flu A and FluB detection and relate this to complications.

The authors also indicated samples were tested using PCR and rapid kits. Authors should indicate the brand and country of origin of the rapid kits and briefly describe the RT-PCR assays used for the testing.

Staff Comments:

Preparing Revision Guidelines

Please return the manuscript within 60 days; if you cannot complete the modification within this time period, please contact me. If you do not wish to modify the manuscript and prefer to submit it to another journal, please notify me of your decision immediately so that the manuscript may be formally withdrawn from consideration by Microbiology Spectrum.

Influenza in Liver and Kidney Transplant Recipients: Incidence and

Outcomes

Nicoline Stender Arentoft MD^a; Dina Leth Møller, MD^a; Andreas Delhbæk Knudsen, MD^a;
Ranya Abdulovski, MD^a; Nikolai Kirkby^b; Søren Schwartz Sørensen, MD, DMsc^{c, d}; Allan
Rasmussen, MD^e; Susanne Dam Nielsen, MD, DMsc^{a, d, e}

a. Viro-immunology Research Unit, Department of Infectious Diseases 8632,
Rigshospitalet, University of Copenhagen, Copenhagen, Denmark.

Nicoline.stender.arentoft.01@regionh.dk (ORCID: 0000-0001-9083-863X)

Dina.leth.moeller@regionh.dk (ORCID: 0000-0002-3909-0643)

Andreas.dehlbaek.knudsen@regionh.dk (ORCID: 0000-0002-5349-7829)

Ranya.houmami.01@regionh.dk (ORCID: 0000-0001-9166-1012)

sdn@dadlnet.dk (ORCID: 0000-0001-6391-7455)

b. Department of Clinical Microbiology, Rigshospitalet, University of Copenhagen,
Copenhagen, Denmark.

Nikolai.kirkby@regionh.dk (ORCID 0000-0001-7498-2738)

c. Department of Nephrology, Rigshospitalet, University of Copenhagen,
Copenhagen, Denmark.

Soeren.Schwartz.Soerensen@regionh.dk (ORCID: 0000-0001-5898-8048)

19 d. Department of Clinical Medicine, University of Copenhagen, Copenhagen,
Denmark.

e. Department of Surgical Gastroenterology and Transplantation, Rigshospitalet,
University of Copenhagen, Copenhagen, Denmark.

Allan.rasmussen@dadlnet.dk (ORCID: 0000-0002-9550-4767)

Author contributions: NSA, DLM, SSS, ADK, AR, and SDN designed the study. All authors
collected the data. NSA and DLM did statistical analyses. NSA, DLM, and SDN wrote the
manuscript. All authors revised and commented on the manuscript. All authors read and
approved the final version of the manuscript.

Funding: This work received funding from the Novo Nordic Foundation.

Conflict of interest: NSA, DLM, RA, NK, SSS and AR did not report any conflict of interest. ADK
received a grant from The Danish Heart Foundation and a grant from the European
Commission not related to this work. SDN received a grant from the Novo Nordic Foundation.

**Abstract**

**Background:** Influenza is a common respiratory tract infection in solid organ transplant
(SOT) recipients. We aimed to investigate the incidence, risk factors, and complications of
influenza in a large cohort of kidney and liver transplant recipients over ten consecutive
seasons.

**Methods:** We conducted a retrospective study including 378 liver and 683 kidney
transplant recipients transplanted from January 1st, 2010, to October 1st, 2019. Data on
influenza were retrieved from MiBa, a nationwide database containing all microbiology
results in Denmark. Clinical data were retrieved from patient records. Incidence rates and
cumulative incidences were calculated, and risk factors were investigated using time-
updated Cox proportional hazards models.

**Results:** The cumulative incidence of influenza in the first 5 years post-transplantation was
6.3% (95% CI: 4.7-7.9%). Of the influenza positive recipients, 65.5% were treated with
oseltamivir, 65.5% were hospitalised, and 16.7% developed pneumonia.

We found no significant effect of same-season influenza vaccination, sex, age or
comorbidities on the risk of acquiring influenza.

**Conclusion:** Incidence of influenza in kidney and liver recipients is high, and 65.5% of
infected transplant recipients required hospitalisation. We were not able to confirm a
reduction in influenza incidence or risk of complications associated with vaccination.

**Importance**

Influenza is a common respiratory virus in solid organ transplant recipients that may have
severe complications, including pneumonia and hospitalization. This study investigates the
incidence, risk factors, and complications of influenza in a Danish cohort of kidney and

liver transplant recipients over ten consecutive influenza seasons. The study shows a high
incidence of influenza and a high frequency of both pneumonia and hospitalization. This
emphasizes the importance of continuous focus on influenza in this vulnerable group.
During the COVID-19 pandemic, the incidence of influenza has been low due to COVID-
related restrictions, and immunity may have waned. However, as most countries have now
opened up, the incidence of influenza is expected to be high this season.

**Introduction**

Due to the immunosuppressive therapy, solid organ transplant (SOT) recipients are at
increased risk of infections compared to immunocompetent individuals (1). Influenza is a
common respiratory infection in kidney and liver recipients, and kidney transplant
recipients have a five-fold higher risk of influenza compared to the background population
(2,3).

SOT recipients with influenza may have a clinical presentation similar to
immunocompetent individuals (4). However, influenza often has a more aggressive course
in SOT recipients, and complications, including pneumonia and bacterial or fungal co-
infections, are common (5,6). Hospitalisation due to influenza in kidney transplant
recipients is four-fold higher compared to the general population, and a mortality reaching
5% has been reported (3).

In SOT recipients seasonal influenza vaccination entails a lower risk of complications, and
annual influenza vaccination is recommended (7,8). However, adherence to the
vaccination programs in SOT recipients is often low with a coverage of 38%-51% (7,9,10).
Importantly, early initiation of antiviral therapy is associated with a lower risk of
complications (4). In contrast, older age, recent use of high-dose corticosteroid and recent
episodes of rejection have been suggested to be risk factors associated with complications
to influenza in SOT recipients (3,5).

In this study of a large cohort of kidney and liver transplant recipients in Denmark, we
investigated the incidence of influenza in ten consecutive seasons from season 2010/2011
to 2019/2020. Furthermore, we describe risk factors associated with acquiring influenza as
well as complications related to influenza.

**Materials and methods**

**Study design and participants**

We present data from *The Knowledge Center for Transplantation* database at
Rigshospitalet, Copenhagen, Denmark. The database includes all first-time liver and
kidney transplant recipients transplanted at Rigshospitalet from January 1st, 2010, to
October 1st, 2019. All liver transplantations in Denmark are performed at Rigshospitalet,
while Denmark has 3 centres for kidney transplantation. The end of follow-up was October
1st, 2020, allowing for at least 1 year of follow-up for all recipients.

The Danish citizens are registered using civil registration (CPR) number. A CPR number is
a unique personal number. All data were collected retrospectively from patient records and
national databases. Data on influenza results were collected from the Danish microbiology
database (MiBa), which contains data on all microbiology from all Departments of Clinical
Microbiology in Denmark with complete coverage since 2010 (11). Information about
influenza vaccination was retrieved from the national Danish Vaccination Registry (DDV),
containing all vaccination data from primary care, hospitals, and vaccination clinics. On
November 15th, 2015, it became mandatory for health care professionals to register all
vaccinations in DDV. Furthermore, the registry contains voluntary vaccination information
from before 2015. We included information about influenza vaccinations administered both
before and after November 15th, 2015. The influenza vaccines used in Denmark are
inactivated and contain 3 or 4 strains. The used strains were as recommended by the
WHO (supplementary table 1) (12). We collected pre-transplantation characteristics,
including age at transplantation, sex, date of transplantation, and comorbidities at the time
of transplantation (diabetes mellitus type I and II, hypertension, chronic heart failure,

ischemic heart disease, chronic obstructive pulmonary disease, and asthma). The post-
transplant characteristics collected were acute organ rejections, influenza vaccination
status, rejection treatment, and influenza test results. We collected information about
influenza-specific outcomes, including antiviral treatment, pneumonia, hospital admission,
admission to an intensive care unit (ICU), mechanical ventilation, and death.

Retrieval of data was approved by the Centre for Regional Development (R-20051155).
Requirement for approval by an ethics committee was waived in accordance with Danish
law, which stipulates that informed consent or approval by an ethics committee is not
needed for studies based on retrospective anonymised data.

**Laboratory influenza analysis**

All ten national Departments of Clinical Microbiology in Denmark contributed to the Miba
database during the study period. Generally, samples were collected as oropharyngeal- or
nasopharyngeal swabs or a combination of both. Other sample materials used for
influenza testing are bronchoalveolar lavage fluid and nasopharyngeal/nasal aspirate.

Samples were analyzed by real-time RT-PCR analysis using in-house or commercial tests
at central lab or Point-Of-Care test in the clinical wards. Multiplex tests including at least
influenza A, influenza B, and internal control were used at all sites.

**Variable definitions**

One influenza season was defined as the period from October 1st to April 30th in two
consecutive calendar years (e.g., October 1st, 2010 to April 30th, 2011), resulting in a

maximum of 211 days at risk per influenza season. Each recipient was at risk of influenza
infection in each influenza season post-transplantation. Having influenza was defined as a
positive PCR test for influenza A or B.

In liver transplant recipients, acute rejection was defined as acute rejection treated with 1 g
of methylprednisolone for three-five days. In kidney transplant recipients, acute rejection
was defined as acute rejection treated with 250 mg or 500 mg of methylprednisolone for
three days.

**Outcome definitions**

The investigated outcomes were antiviral treatment, pneumonia, hospitalisation, ICU
admission, mechanical ventilation, and death. Antiviral treatment was defined as treatment
with the neuraminidase inhibitor oseltamivir initiated seven days before to seven days after
confirmed influenza infection. Pneumonia was defined as infiltrates on chest x-ray together
with a radiology report consistent with pneumonia. Hospitalisation, ICU admission, and
mechanical ventilation five days prior to influenza infection and up to 30 days after
infection were included as outcomes of influenza. These outcomes were assessed only for
individuals with influenza infection.

**Statistical analyses**

Comparisons were calculated using the Mann-Whitney U test for continuous variables and
the Chi² test or Fisher's exact test for categorical variables. The incidence rates (IR) of the
first influenza infection per season were calculated as the number of recipients with an

influenza infection per person month at risk for each influenza season. We calculated 95%
confidence intervals (CI) using Byar's approximation to the Poisson distribution. The 5-year
cumulative incidence of influenza post-transplantation was calculated using the Aalen-
Johansen estimator, with death and re-transplantation as competing risks. We used Gray's
test to compare cumulative incidences between kidney and liver transplanted recipients
(13). Risk factors for influenza infection were investigated in a time-updated multivariable
Cox proportional hazards model with acute rejection and vaccination as time-updated
covariates. The model was adjusted for age, sex, and pre-transplant comorbidities. All
analyses were conducted using R statistical software version 3.6.1 with the (survival,
ggplot2, cmprsk) packages (14).

**Results**

Between January 1st, 2010 and October 1st, 2019, 683 kidney and 378 liver transplant
recipients were included in the study (Table 1). The median age at the time of
transplantation was 51 years (range 18-84), and 61% of the cohort were men. The
distribution of comorbidities is shown in Table 1. A total of 386 episodes of acute rejection
were observed in 294 recipients (27.7%).

**Influenza vaccination**

A total of 713 recipients (67%) were influenza vaccinated in at least one of the 10 seasons,
and 124 (11,7%), 100 (9,4%), and 34 (3,2%) recipients were vaccinated in two, three, and
all seasons, respectively. Only 19 (22.6%) received influenza vaccination in the same
season as their influenza infection. The distribution of vaccinated recipients was
comparable in the two organ groups (69.5% vs 63.0%).

**Influenza**

The median follow-up was 5.6 years (IQ range 3.4-8.3), with a total follow-up of 6152
person-years in the cohort. Eighty-one recipients (7.6%) were diagnosed with influenza
during the follow-up. Of these, three recipients had influenza twice in different seasons,
resulting in a total of 84 influenza infections in the cohort. The median time to infection was
824 days post-transplantation (IQR 251-1930 days). There were significantly more
influenza infections in the kidney recipients than in liver recipients (79.0% vs. 63.1%, p
=0.008). There were no other significant differences in clinical characteristics or number of
influenza vaccinations in any season between the recipients with influenza compared to
the recipients without influenza (Table 2).

**Influenza incidence rate and cumulative incidence**

The incidence rate of influenza in each season from season 2010/2011 to season
2019/2020 ranged between 0 (0.0-1.6) and 5.4 (95% CI 1.8-12.9) per 1000 person-months
at risk (supplementary Figure 1 and supplementary Table 2). There were no significant
differences in the incidence rates between the recipients who had received same-season
influenza vaccination and recipients who had not (supplementary Table 2).

The cumulative incidence of first influenza infection in the first year and first five years
post-transplantation was 2.5% (95% CI: 1.6-3.5%) and 6.3% (95% CI: 4.7-7.9%),
respectively (Figure 1). The kidney transplant recipients had higher cumulative incidence
of influenza than liver recipients (7.7% (95% CI 5.5-9.8%) vs 3.7% (95% CI 1.6-5.8%), p =
0.02).

**Risk factors for influenza**

Risk factors associated with being diagnosed with influenza were tested in a time-updated
Cox proportional hazards model. Age, sex, having at least one comorbid disease and
same-season influenza vaccination were not associated with the risk of influenza (Table
3). In unadjusted and adjusted analyses, the risk of acquiring influenza had a hazard ratio
for same-season vaccination of 1.3 (95% CI 0.8-2.2, p=0.3) and 1.4 (95% CI 0.8-2.3,
p=0.2), respectively.

**Outcome after influenza infection**

Of the 84 influenza infections, 66 (78.6%) and 18 (21.4%) were found in kidney and liver
recipients, respectively (Table 4). Of the influenza positive recipients, 55 of the 84 (65.5%)
were treated with oseltamivir, 55 (65.5%) were hospitalised, and 14 recipients (16.7%)
developed pneumonia. Six (7.1%) recipients were admitted to an ICU and had supportive
mechanical ventilation within 30 days of their influenza infection. Only one (1.2%) recipient
died within 30 days of positive influenza PCR. There were no significant differences in
outcomes when comparing kidney and liver recipients (Table 4) nor did we find any
difference in outcomes when comparing influenza positive recipients, who had received
same-season vaccination with those who had not (supplementary Table 3). Pneumonia
was numerically more common in the recipients who had not been vaccinated in the same
season, but the finding was not significant (10.5% vs 18.5%, p=0.7).

**Discussion**

This retrospective cohort study of kidney and liver transplant recipients investigated the
cumulative incidence of influenza in the first five years post-transplantation, showing a
cumulative incidence that continues to rise over time and a higher incidence in kidney
transplant recipients compared to liver transplant recipients. Complications and risk factors
associated with influenza were studied with no significant effect of vaccination in the same
season as influenza and no association with any other potential risk factors.

Hospitalisation and pneumonia were common complications to influenza in both kidney
and liver recipients and the frequency was not lower in recipients who had received same-
season vaccination.

We found a cumulative incidence of influenza of 2.5% and 6.3% in the first year and the
first five years post-transplantation, respectively. A recent Swiss study including 3294 SOT
recipients found a similar cumulative incidence of influenza in non-lung transplant
recipients of 7.5% in the first 7.5 years post-transplantation (2). In general, SOT recipients
are most susceptible to infections in the early post-transplantation phase due to the high
immunosuppression (1). However, the cumulative incidences in our study showed
influenza in both early and later post-transplantation periods which may be explained by
influenza being a community-acquired viral infection that poses a persistent risk of
infection every year in the influenza season, unrelated to immunosuppression. These
results highlight the importance of continuous awareness of influenza post-transplantation.

Our study showed an incidence rate between 0 and 5.4 per 1000 person months in ten
consecutive influenza seasons. An American study, including 3569 SOT recipients, found
a similar incidence rate of influenza varying between 2.8-4.3 per 1000 person-years in

lung, liver, and kidney transplant recipients (15). This highlights the importance of including
multiple influenza seasons when investigating influenza epidemiology. We found zero
influenza events in the 2011/2012 season, in accordance with the general population in
Denmark, where the number of patients with influenza-like symptoms was at the lowest
level since surveillance started in 1994 (16). Likewise, the high incidence rate of influenza
in the season 2017/2018 reflected the unusually long influenza season with high activity in
the general population in Denmark (17).

In contrast to our expectations, same-season influenza vaccination was not associated
with lower risk of acquiring influenza. Neither did we find evidence of fewer complications
to influenza in recipients who had received same-season vaccination. There could be
several possible reasons for this observation. First, studies of standard-dose influenza
vaccination have shown lower overall antibody response in SOT compared to healthy
controls (7). Furthermore, the strains in the yearly vaccine may not protect against the
circulating strains (12). Lastly, our cohort may have been underpowered due to the varying
protection of the yearly vaccination. However, annual vaccination is recommended since
influenza vaccination is considered safe in SOT recipients (4,7), and vaccination has been
associated with less severe disease in several studies including a recent Danish study
from our group which found reduced risk of hospitalisation and mortality in vaccinated
recipients following influenza infection (8). Our study did not show an association between
age, sex, comorbidities, and risk of influenza. Only few other studies of risk factors of
influenza in SOT have been done, including a study from Finland of kidney transplant
recipients that found the time periode in which the transplantation had been performed, to
be the only significant risk factor (3).

Our study found that influenza infection was more common in kidney transplant recipients
than in liver transplant recipients, but there was no difference in complications associated
with influenza infection between the organ groups. Hospitalisation was seen in 65.5% of
kidney and liver transplant recipients with influenza and 7.1% was admitted to an ICU. In a
recent study from Canada of 443 SOT recipients with influenza, 69.3% of the cohort were
hospitalised, and 7.8% needed mechanical ventilation (4). Furthermore, a Swiss study of
186 influenza infections in SOT recipients found that 42.9% of the patients needed hospital
admission and 4.3% required mechanical ventilation (2). In Finland , kidney recipients
have a fourfold higher risk of hospitalisation due to influenza compared to the general
population emphasising the increased risk associated to influenza in SOT recipients (3).

The strengths of our study include the large well described cohort with a long, complete
follow-up across ten consecutive influenza seasons based on Scandinavian registers as
well as complete coverage of influenza test results. However, our study also had possible
limitations. First, information on vaccination became mandatory on November 15th, 2015,
and some vaccinations administered before this date may not be included in our study.
Our database does not include data on immunosuppression, apart from treatment for
rejection. Despite our complete coverage of influenza PCR results in Denmark, there is no
routine influenza testing of transplant recipients. Patients with mild symptoms, such as
vaccinated patients, are therefore unlikely to be tested, and this patient group may be
underrepresented in this study. Lastly, the number of influenza infections in our cohort
limits the power of our statistical analyses.

In summary, this study provides new knowledge on the epidemiology of influenza in kidney
and liver transplant recipients. Incidence rates of influenza in kidney and liver recipients

fluctuates, following the influenza activity in the general population. The cumulative
incidence in our study demonstrates the importance of continuous awareness of influenza
in kidney and liver recipients. Influenza often leads to pneumonia and hospitalisation in
kidney and liver transplant recipients, highlighting the severe consequences of influenza in
these groups. We were not able to confirm a reduction in risk of influenza or risk of
complications associated with same-season vaccination.

**References**

- 1. Fishman JA. Infection in solid-organ transplant recipients. *N Engl J Med*. 2007
Dec;357(25):2601–14. DOI10.1056/NEJMra064928
- 2. Mombelli M, Lang BM, Neofytos D, Aubert JD, Benden C, Berger C, et al. Burden,
epidemiology, and outcomes of microbiologically confirmed respiratory viral
infections in solid organ transplant recipients: a nationwide, multi-season prospective
cohort study. *Am J Transplant Off J Am Soc Transplant Am Soc Transpl Surg*.
2020 Oct; DOI10.1111/ajt.16383
- 3. Helanterä I, Gissler M, Rimhanen-Finne R, Ikonen N, Kanerva M, Lempinen M, et al.
Epidemiology of laboratory-confirmed influenza among kidney transplant recipients
compared to the general population-A nationwide cohort study. *Am J Transplant Off*
*J Am Soc Transplant Am Soc Transpl Surg*. 2021 May;21(5):1848–56.
DOI10.1111/ajt.16421
- 4. Kumar D, Ferreira VH, Blumberg E, Silveira F, Cordero E, Perez-Romero P, et al. A
5-Year Prospective Multicenter Evaluation of Influenza Infection in Transplant
Recipients. *Clin Infect Dis an Off Publ Infect Dis Soc Am*. 2018 Oct;67(9):1322–9.
DOI10.1093/cid/ciy294
- 5. Abbas S, Raybould JE, Sastry S, de la Cruz O. Respiratory viruses in transplant
recipients: more than just a cold. *Clinical syndromes and infection prevention*
*principles. Int J Infect Dis IJID Off Publ Int Soc Infect Dis*. 2017 Sep;62:86–93.
DOI10.1016/j.ijid.2017.07.011
- 6. Manuel O, López-Medrano F, Keiser L, Welte T, Carratalà J, Cordero E, et al.
Influenza and other respiratory virus infections in solid organ transplant recipients.
*Clin Microbiol Infect Off Publ Eur Soc Clin Microbiol Infect Dis*. 2014 Sep;20 Suppl

- 7(Suppl 7):102–8. DOI10.1111/1469-0691.12595
- 7. Danziger-Isakov L, Kumar D. Vaccination of solid organ transplant candidates and
recipients: Guidelines from the American society of transplantation infectious
diseases community of practice. *Clin Transplant*. 2019 Sep;33(9):e13563.
DOI10.1111/ctr.13563
- 8. Harboe ZB, Modin D, Gustafsson F, Perch M, Gislason G, Sørensen SS, et al. Effect
of influenza vaccination in solid organ transplant recipients: A nationwide population-
based cohort study. *Am J Transplant [Internet]*. 2022 May 16; Available from:
<https://onlinelibrary.wiley.com/doi/10.1111/ajt.17055> DOI10.1111/ajt.17055
- 9. Larsen L, Bistrup C, Sørensen SS, Boesby L, Nguyen MTT, Johansen IS. The
Coverage of Influenza and Pneumococcal Vaccination among Kidney Transplant
Recipients and Waiting List Patients: A Cross Sectional Survey in Denmark. *Transpl
Infect Dis*. 2020 Nov;e13537. DOI10.1111/tid.13537
- 10. Hirzel C, Kumar D. Influenza vaccine strategies for solid organ transplant recipients.
*Curr Opin Infect Dis*. 2018 Aug;31(4):309–15.
DOI10.1097/QCO.0000000000000461
- 11. Voldstedlund M, Haarh M, Mølbak K. The Danish Microbiology Database (MiBa)
2010 to 2013. *Euro Surveill Bull Eur sur les Mal Transm = Eur Commun Dis Bull*.
2014 Jan;19(1). DOI10.2807/1560-7917.es2014.19.1.20667
- 12. Statens Serum Institut. Influenzavaccination [Internet]. 2020 [cited 2020 Dec 9].
Available from: <https://www.ssi.dk/vaccinationer/influenzavaccination>
- 13. Gray RJ. A Class of K-Sample Tests for Comparing the Cumulative Incidence of a
Competing Risk. *Ann Stat [Internet]*. 1988 Jun 15;16(3):1141–54. Available from:
<http://www.jstor.org/stable/2241622>

- 14. R Development Core Team R. R: A Language and Environment for Statistical
Computing. R Foundation for Statistical Computing. 2011. DOI10.1007/978-3-540-
74686-7
- 15. Vilchez RA, McCurry K, Dauber J, Lacono A, Griffith B, Fung J, et al. Influenza virus
infection in adult solid organ transplant recipients. Am J Transplant Off J Am Soc
Transplant Am Soc Transpl Surg. 2002 Mar;2(3):287–91. DOI10.1034/j.1600-
6143.2002.20315.x
- 16. Statens Serum Institut. EPI-NYT: Influenza Season 2011/2012 [Internet]. 2012 [cited
2020 Dec 9]. Available from: [https://www.ssi.dk/aktuelt/nyhedsbreve/epi-
nyt/2012/uge-24a---2012](https://www.ssi.dk/aktuelt/nyhedsbreve/epi-nyt/2012/uge-24a---2012)
- 17. Statens Serum Institut. EPI-NYT: Influenza Season 2017/2018. [Internet]. 2018
[cited 2020 Dec 4]. Available from: [https://www.ssi.dk/aktuelt/nyhedsbreve/epi-
nyt/2018/uge-23_24---2018](https://www.ssi.dk/aktuelt/nyhedsbreve/epi-nyt/2018/uge-23_24---2018)

Organ	681	639	629	592	551	518	470	431	386	361	323
Kidney:	681	639	629	592	551	518	470	431	386	361	323
Liver:	378	334	320	295	270	249	230	206	185	167	144

Table 1			
Characteristics	All (n = 1061)	Kidney transplanted recipients (n = 683)	Liver transplanted recipients (n = 378)
Age at transplantation, year, median (range)	50.8 (18.0-83.5)	50.9 (19.0-83.5)	50.2 (18.0-73.8)
Male sex, n (%)	646 (61.0%)	428 (62.7%)	218 (57.7%)
Patients with ≥1 comorbidity, n (%)*			
- Diabetes mellitus type I or II, n (%)	180 (17.0%)	121 (17.7%)	59 (15.6%)
- Cardiovascular disease, n (%)	689 (64.9%)	611 (89.5%)	78 (20.6%)
- Chronic lung disease, n (%)	102 (9.6%)	66 (9.7%)	36 (9.5%)
Dead, n (%)	212 (20.0%)	134(19.6%)	78 (20.6%)
Rejection, n (%)	284 (26.8%)	206 (30.2%)	78 (20.6%)
Influenza vaccinated in any season, n (%)	713 (67.2%)	475 (69.5%)	238 (63.0%)
- Influenza vaccinated in 2010, n (%)	141 (13.3%)	97 (14.2%)	44 (11.6%)
- Influenza vaccinated in 2011, n (%)	161 (15.2%)	112 (16.4%)	49 (13.0%)
- Influenza vaccinated in 2012, n (%)	175 (16.5%)	121 (17.7%)	54 (14.3%)
- Influenza vaccinated in 2013, n (%)	212 (20.0%)	139 (20.4%)	73 (19.3%)
- Influenza vaccinated in 2014, n (%)	192 (18.1%)	128 (18.7%)	64 (16.9%)
- Influenza vaccinated in 2015, n (%)	235 (22.1%)	155 (22.7%)	80 (21.2%)
- Influenza vaccinated in 2016, n (%)	306 (28.8%)	196 (28.7%)	110 (29.1%)
- Influenza vaccinated in 2017, n (%)	331 (31.2%)	213 (31.2%)	118 (31.2%)
- Influenza vaccinated in 2018, n (%)	370 (34.9%)	237 (34.7%)	133 (35.2%)
- Influenza vaccinated in 2019, n (%)	370 (34.9%)	234 (34.3%)	136 (36.0%)

*comorbidities at time of transplantation

Table 2 influenza positive recipients vs influenza negative recipients			
Characteristics	Influenza positive recipients*	Influenza negative recipients	P-values
Number of patients, n	81	980	
Age at transplantation, year, median (range)	49.7 (19.4-71.2)	50.9 (18.0-83.5)	P=0.6
Male sex, n (%)	46 (56.8%)	600 (61.3%)	P=0.6
Transplanted organ			
- Kidney	64 (79.0%)	617 (63.1%)	P=0.008
- Liver	17 (21.0%)	361 (36.9%)	
Influenza vaccination in any season	56 (69.1%)	655 (67.0%)	P=0.9
Patients with ≥1 comorbidity, n (%)			
- Diabetes mellitus type I and II, n (%)	14 (17.3%)	166 (17.0%)	P=0.9
- Chronic heart disease, n (%)	61 (75.3%)	626 (64.0%)	P=0.1
- Chronic lung disease, n (%)	9 (11.1%)	93 (9.5%)	P=0.9
Rejection, n (%)	27 (33.3%)	255 (26.1%)	P=0.2
Dead, n (%)	17 (21.0%)	194 (19.8%)	P=0.9
*3 recipients had influenza in two different seasons			

Table 3 Risk factors			
Risk factor	Unadjusted (95% CI)	adjusted for age and sex (95% CI)	Adjusted for age, sex and having ≥ 1 comorbiditet (95% CI)
Age	1.0 (0.98-1.0), p=0.8	1.0 (0.98-1.0), p=0.8	1.0 (0.98-1.0), p=0.8
Sex	0.9 (0.6-1.3), p=0.5	0.9 (0.6-1.3), p=0.5	0.9 (0.6-1.3), p=0.5
Comorbidity	1.4 (0.8-2.6) p=0.3	1.5 (0.8-2.7) p=0.2	1.5 (0.8-2.7) p=0.2
Same-season influenza vaccine	1.3 (0.8-2.2), p=0.3	1.4 (0.8-2.2), p=0.2	1.4 (0.8-2.3) p=0.2

Table 4 Outcomes after influenza infection in kidney and liver transplanted recipients

	All Influenza infections (n =84)	Influenza infections in kidney transplanted recipients (n=66 (79%))	Influenza infections in liver transplanted recipients (n=18 (21%))	P-values
Time from tx to influenza, days, median (IQ range)	824 (251.3-1929.8)	792 (209.5-2059)	1002.5 (345.3-1769.3)	P=1
Same-season influenza vaccination	19 (22.6%)	17 (25.8%)	2 (11.1%)	P=0.4
Treated with oseltamivir (%)	55 (65.5%)	45 (68.2%)	10 (55.6%)	P=0.7
Pneumonia (%)	14 (16.7%)	13 (19.7%)	1 (5.6%)	P=0.3
Hospital admission (%)	55 (65.5%)	44 (66.7%)	11 (61.1%)	P=1
ICU admission (%)	6 (7.1%)	5 (7.6%)	1 (5.6%)	P=1
Mechanical ventilation (%)	5 (6.0%)	4 (6.1%)	1 (5.6%)	P=1
Death, 30 days all-cause mortality (%)	1 (1.2%)	1 (1.5%)	0 (0%)	P=1
Influenza type (%)				
- A	53 (63.1%)	44 (66.7%)	9 (50%)	P=0.7
- B	31 (36.9%)	22 (33.3%)	9 (50%)	P=0.5

Point-by-point answer

Dear Editor,

Thank you for allowing us to revise and improve our manuscript. We thank the reviewer for the excellent comments. We have revised the manuscript according to the comments, and in doing so, we believe the quality of the manuscript has improved. Below is a point-by-point reply to the reviewers' comments. All references to the line and page numbers are to the manuscript version with "Track changes."

Comments from the reviewer

Influenza is a respiratory virus that is associated with morbidity and mortality across the world. The incidence, risk factors and complications among cohorts of kidney and liver transplant recipients has not been extensively studied.

This study is therefore useful and likely to inform policy and stakeholders in making key decisions about these infections.

Authors should respond to the comments below:

1. Line 58: Authors should state the number of positives relating to the second sentence. Eg:
"Of the xxx influenza positive recipients, 65.5% were

Reply

Thank you for the comment. The sentence on line 58 has been changed to:

"Of the 84 influenza positive recipients, 65.5% were.."

2. Line 108: "Citisens" should be "citizens"

Reply

Thank you for noticing this typo, which have now been corrected.

3. Line 225-227: Authors should recheck the statement again. It appears positive cases were 84. Fourteen (14) recipients developed pneumonia so were the rest asymptomatic?

Reply

Thank you for this comment. It is correct that we found 84 influenza cases and of these 14 had pneumonia. Below is a table showing symptoms for all 84 cases. This has been added to the supplementary document.

On line 230-231 the following has been added:

“The clinical symptoms of the remaining 70 recipients (83.3%) in relation to influenza can be seen in supplementary Table 3.”

Transplanted organ	Influenza type	Symptoms in relation to influenza	Pneumonia	Hospital	ICU	Ventilator
Liver	A	Fever, muscle pain	No	Yes	No	No
Liver	B	Fever, diarrhea, vomiting	No	Yes	No	No
Liver	B	Bronchitis symptoms	No	No	No	No
Liver	B	Fever	No	Yes	No	No
Liver	B	Fever, vomiting, muscle pain, shortness of breath	No	Yes	No	No
Liver	A	Fever, muscle pain	No	Yes	No	No
Liver	B	Fever	No	No	No	No
Liver	A	Fever, cough, headache, muscle pain	No	No	No	No
Liver	A	Pneumonia	Yes	Yes	Yes	Yes
Liver	B	Shortness of breath, muscle pain, headache, vomiting	No	No	No	No
Liver	A	Cough, throat pain	No	No	No	No
Liver	A	Fever, cough	No	Yes	No	No
Liver	B	Fever, abdominal pain, headache, cough	No	No	No	No
Liver	B	Unknown	No	No	No	No
Liver	A	Fever	No	Yes	No	No
Liver	A	Fever, headache, cough	No	Yes	No	No
Liver	B	Fever, headache, cough, joint pain	No	Yes	No	No
Liver	A	Fever, cough, muscle pain	No	No	No	No
Kidney	A	Fever, vomiting, cough	No	Yes	No	No

Kidney	B	Fever, cough	No	Yes	No	No
Kidney	A	Fever, cough	No	No	No	No
Kidney	A	Fever, cough, muscle- and joint pain	No	Yes	No	No
Kidney	A	Fever, nausea	No	Yes	No	No
Kidney	B	Fever, cough	No	Yes	No	No
Kidney	B	Pneumonia	Yes	Yes	No	No
Kidney	A	Decline in graft function, generally uncomfortable	No	Yes	No	No
Kidney	A	Fever, cough	No	Yes	No	No
Kidney	A	Fever, cough	No	Yes	No	No
Kidney	B	Headache, cough	No	Yes	No	No
Kidney	A	Pneumonia	Yes	Yes	No	No
Kidney	B	Tired	No	No	No	No
Kidney	A	Fever	No	Yes	No	No
Kidney	A	Unspecified influenza symptoms	No	No	No	No
Kidney	B	Fever, cough	No	Yes	No	No
Kidney	B	Vomiting, diarrhea, throat pain, respiratory insufficient	No	Yes	Yes	Yes
Kidney	A	Fever, cough	No	No	No	No
Kidney	B	Fever, cough	No	No	No	No
Kidney	A	Fever, cough, muscle- and joint pain, headache	No	Yes	No	No
Kidney	A	Pneumonia	Yes	Yes	No	No
Kidney	B	Fever, cough	No	Yes	No	No
Kidney	A	Fever	No	No	No	No
Kidney	A	Fever, respiratory insufficient	No	Yes	No	No
Kidney	A	Fever, cough, headache, tired	No	Yes	No	No
Kidney	A	Fever, respiratory insufficient	No	Yes	Yes	Yes
Kidney	B	Fever, cough, respiratory insufficient	No	Yes	No	No
Kidney	A	Fever, cough, joint pain	No	Yes	No	No
Kidney	A	Fever, muscle pain, cough	No	Yes	No	No
Kidney	A	Cough, respiratory insufficient	No	No	No	No
Kidney	B	Fever, nausea	No	No	No	No
Kidney	B	Pneumonia	Yes	Yes	No	No
Kidney	B	Fever, cough	No	No	No	No
Kidney	B	Pneumonia	Yes	Yes	Yes	Yes
Kidney	A	Pneumonia	Yes	Yes	No	No
Kidney	A	Fever, cough, nausea, headache	No	No	No	No
Kidney	B	Pneumonia	Yes	No	No	No

Kidney	A	Fever, cough	No	No	No	No
Kidney	B	Fever, cough, headache	No	No	No	No
Kidney	A	Asymptomatic	No	Yes	No	No
Kidney	A	Pneumonia	Yes	Yes	No	No
Kidney	A	Fever, vomiting, cough	No	Yes	No	No
Kidney	A	Dyspnea	No	Yes	No	No
Kidney	A	Fever, cough	No	No	No	No
Kidney	B	General discomfort	No	Unknown	No	No
Kidney	A	Fever, headache	No	Yes	No	No
Kidney	A	Pneumonia	Yes	Yes	No	No
Kidney	A	Unknown	No	No	No	No
Kidney	A	Cough	No	No	No	No
Kidney	A	Fever, headache, cough	No	Yes	No	No
Kidney	B	Fever, headache, muscle pain	No	No	No	No
Kidney	A	Pneumonia	Yes	Yes	No	No
Kidney	A	Pneumonia	Yes	No	No	No
Kidney	A	Fever, cough	No	No	No	No
Kidney	A	Fever, cough	No	Yes	No	No
Kidney	B	Sore throat	No	No	No	No
Kidney	A	Vomiting, cough	No	Yes	No	no
Kidney	A	Pneumonia	Yes	Yes	Yes	Yes
Kidney	A	Pneumonia	Yes	Yes	Yes	No
Kidney	A	Fever, cough	No	Yes	No	No
Kidney	A	Cough, abdominal pain, nausea, vomiting	No	Yes	No	No
Kidney	A	Cough	No	No	No	No
Kidney	B	Fever, cough	No	Yes	No	No
Kidney	B	Fever, diarrhea,	No	Yes	No	No
Kidney	A	Fever, headache, sore throat	No	Yes	No	No
Kidney	A	Fever, headache, cough, sore throat	No	Yes	No	No

4. Line 296-298: Authors mentioned that some vaccinations administered could not be included. Authors should indicate the number of number of vaccinations that were not included.

Reply

Thank you for this comment. As mentioned in the text line 115-120 it has been mandatory for all healthcare workers to register all vaccines administered in Denmark since November 15th, 2015. Before this date it was voluntary to register vaccines. Therefore, we do not know the exact number

of vaccines administered before November 15th, 2015. The table below shows administered vaccines for each season and the number recipients in each season. Based on this table it does not seem like fewer vaccines were registered before 2015.

Line 304 in the main text have been changed to

"Information on vaccination became mandatory on November 15th, 2015, but when comparing vaccination numbers across seasons in our cohort, it does not seem like fewer vaccines were registered before 2015."

	Recipients in season, n	Influenza vaccinated recipients in season, n (%)
142	141 (99.3%)
253	161 (63.6%)
326	175 (53.7%)
420	212 (50.5%)
510	192 (37.6%)
621	235 (37.8%)
718	306 (42.6%)
822	331 (40.3%)
891	370 (41.5%)
923	370 (40.1%)

5. General comments

Table 4: Although authors included flu A and B results, this was not described in the text.

Flu A is much more severe than Flu B so it will be interesting to discuss this. The abstract should as well capture Flu A and FluB detection and relate this to complications.

Reply

Thank you very much for the comment. We certainly agree that influenza A is often more severe than influenza B. We have compared the complications between the two groups, shown in the table below. Although there are no significant differences between the groups, it looks like patients

with influenza A were admitted to the hospital more often than patients with influenza B. This table will be added as supplementary table 4.

Line 58-61 in the abstract has been changed to:

“Of the 84 influenza positive recipients, 63.1% had influenza A, 65.5% were treated with oseltamivir, 65.5% were hospitalised, and 16.7% developed pneumonia. There was no significant difference on outcomes when comparing patients with influenza A and B.”

On line 239-241 in the text, the following has been added:

“When comparing outcomes for patients with influenza A and influenza B there were no significant differences, although hospitalization tended to be more common in patients with influenza A than in patients with influenza B (73.6% vs 51.6%, $p=0.07$) (supplementary Table 5).”

	All	Influenza A	Influenza B	P-values
Number of patients in group, n	84	53	31	
Pneumonia, n (%)	14	10 (18.9)	4 (12.9)	P=0.56
Hospital admission, n (%)	55	39 (73.6)	16 (51.6)	P=0.07
ICU Admission, n (%)	6	4 (7.5)	2 (6.5)	P=1
Mechanical ventilation, n (%)	5	3 (5.7)	2 (6.5)	P=1

6. The authors also indicated samples were tested using PCR and rapid kits. Authors should indicate the brand and country of origin of the rapid kits and briefly describe the RT-PCR assays used for the testing.

Reply

Thank you for this comment. We have consulted our Department of Microbiology. Because this study includes tests from a 10-year period from all ten Departments of Microbiology in Denmark, it is not possible to account for all the used Point-Of-Care and RT-PCR tests.

March 9, 2023

Prof. Susanne Dam Nielsen
Copenhagen University Hospital, Rigshospitalet
Viro-immunology Research Unit - Department of Infectious Disease
Blegdamsvej 9
Copenhagen
Denmark

Re: Spectrum03226-22R1 (Influenza in Liver and Kidney Transplant Recipients: Incidence and Outcomes)

Dear Prof. Susanne Dam Nielsen:

Your manuscript has been accepted, and I am forwarding it to the ASM Journals Department for publication. I apologize for the delay as I was dealing with personal issues. You will be notified when your proofs are ready to be viewed.

Sincerely,

Daniel Perez
Editor, Microbiology Spectrum
